

# *SoilErosionDB*: A global database for surface runoff and soil erosion evaluation

Jinshi Jian[1]★, Xuan Du[2]★, Ryan D. Stewart[3], Zeli Tan[4], Ben Bond-Lamberty[1]

[1]Pacific Northwest National Laboratory, Joint Global Change Research Institute at the University of
Maryland–College Park, 5825 University Research Court, Suite 3500, College Park, MD 20740, USA
[2]Department of Hydraulic Engineering, Yangling Vocational & Technical College, Yang Ling, Shaanxi,
China
[3]School of Plant and Environmental Sciences, Virginia Tech, Blacksburg, VA, USA
[4]Pacific Northwest National Laboratory, Richland, WA, USA

*Correspondence to: Jinshi Jian (jinshi@vt.edu)*

★ These authors contributed equally to this work

## Abstract

Soil erosion is a major threat to soil resources, continuing to cause environmental degradation and social poverty in many parts of the world. Many field and laboratory experiments have been performed over the past century to study spatio-temporal patterns of soil erosion caused by surface runoff under different environmental conditions. However, these historical data have never been integrated together in a way that can inform current and future efforts to understand and model soil erosion at different scales. Here, we designed a database (*SoilErosionDB*) to compile field and laboratory measurements of soil erosion caused by surface runoff, with data coming from sites across the globe. The *SoilErosionDB* includes 18 columns for soil erosion related indicators and 73 columns for background information that describe factors such as latitude, longitude, climate, elevation, and soil type. Currently, measurements from 99 geographic sites and 22 countries around the world have been compiled into *SoilErosionDB*. We provide examples of linking *SoilErosionDB* with an external climate dataset and using annual precipitation to explain annual soil erosion variability under different environmental conditions.

All data and code to reproduce the results in this study can be found at: Jian, J., Du, X., Stewart, R., Tan, Z. and Bond-Lamberty, B.: jinshijian/SoilErosionDB: First release of SoilErosionDB, Zenodo, doi:10.5281/zenodo.4030875, 2020b. All data are also available through GitHub: https://github.com/jinshijian/SoilErosionDB.

**Keywords**: Surface runoff, soil erosion, nutrient leaching, database



## 1. Background

Soil is an essential natural resource for human sustainable development that is continually threatened by erosion and related land degradation processes (Borrelli et al., 2017; Poesen, 2017). Soil erosion is a geomorphic process that occurs when soil particles, soil aggregates, organic matter, and rock fragments detach from their original positions and become transported to other locations (Morgan, 1988; Toy et al., 2002). Erosion is a naturally occurring process affected by both abiotic (e.g., rainfall, runoff, wind, and snow avalanches) and biotic drivers (e.g., animal trampling and tree fall), and may have significant consequences for carbon cycling (Berhe et al., 2007; Ito, 2007; Lal, 2003; Tan et al., 2020; Yue et al., 2016).

Erosion rates have been substantially increased by human activities such as cropland cultivation, mining, building construction, and deforestation (Borrelli et al., 2017; García-Ruiz et al., 2015; Poesen, 2017; Vanwalleghem et al., 2017). For instance, large-scale farmland development in forested and grass-covered prairies of the United States of America led to the Black Dust Bowl of the 1930s, resulting in widespread ecosystem damage and economic losses (Schubert et al., 2004). In China, population and demographic shifts have increased demand for food and other resources, leading to increases in cultivated croplands and concomitant soil erosion problems (Guo et al., 2015; Xu et al., 2010; Zhang et al., 2012; Zhao et al., 2013).

Many field and lab experiments have characterized soil erosion under different environmental conditions. Some have utilized soil erosion runoff plots, with nations and regions such as the USA (>10,000 plots), Europe (>8.000 plots), Brazil (3,525 plots) and Brazil (>2,500 plots) all having widespread installations (Poesen, 2017). These experiments have led to more than 31,000 different peer-reviewed studies (searched at https://www.sciencedirect.com using "Erosion and runoff" as keyword). However, there has not yet been a successful effort to compile the data from those studies into a single and coherent dataset.

To bridge this research gap, we developed a global soil erosion database (*SoilErosionDB*) for standardizing and compiling historical soil erosion related measurements together. The database can be used to support evaluation and parameterization of global soil erosion models, statistical modeling, non-point pollution evaluation, as well as cropland management recommendation. It can also be used to perform synthesis analyses, such as meta-analyses, and may inspire future efforts to better understand spatial and temporal patterns of soil erosion.





## 2. Methods

We designed the *SoilErosionDB* following FAIR protocols, i.e., Findable, Accessible, Interoperable, and Reusable (Wilkinson et al., 2016). All data, quality assurance/quality control (QA/QC) code, and analysis code are immediately available through a GitHub repository (https://github.com/jinshijian/SoilErosionDB), and each release will be issued a DOI through 

Zenodo to ensure reusability. The version format follows an "*x.y.z*" format, where *x* is the major version number, *y* is the minor version number, and *z* is the patch number. We update the major version number only if the database changes its structure; we expect this to happen at an approximately decadal time step. We update the minor version number whenever the database has a significant data update; this usually happens at annual time steps. The patch number will be

updated relatively often, whenever the database has an important documentation update or data correction. We also made efforts to ensure interoperability so that *SoilErosionDB* could easily link to external datasets. For an example of linking *SoilErosionDB* to an external climate dataset please see "4. **Linkages to external data sources**" section below.

### 2.1 Publication collection

Publications were collected during an online literature search using "runoff, erosion" as keyword in ScienceDirect (https://www.sciencedirect.com/). We had no restrictions on literature types, i.e., both peer-reviewed articles and no-peer-reviewed articles such as theses, dissertations, and conference collections were included. We initiated our search on January 10, 2020 and found 31,235 papers, with an increasing number of published papers by year (**Figure 1a**). The following

criteria were used to determine whether a article should be included in the *SoilErosionDB*: (1) measurements were measured in the field, at the laboratory with rainfall simulation experiment, or from indirect methods (**Table 1**); (2) soil erosion was reported in units that could be converted to t ha$^{-1}$ year$^{-1}$ or g m$^{-2}$ hour$^{-1}$; and (3) articles were published in English or Chinese language journals after 1960. We included no other filtering criteria or restrictions to the literature. Note that we did

not include a constraint for leaching data because a variety of leaching types (e.g., soil organic carbon loss, organic matter losses, and total nitrogen loss) were reported in papers, with those measurements reported in different units.

### 2.2 Database structure design

The *SoilErosionDB* (i.e., "*SoilErosionDB*.xlsx" file in the GitHub repository) has 12 data sheets, 

and the core part is the "*SoilErosionDB*" data sheet, with 18 columns for soil erosion, surface runoff, and nutrient leaching records (**Table 2**) and 73 columns for background information (**Table 3**). The "DataBase_fields" sheet describes all 91 columns in the "*SoilErosionDB*" data sheet. The



"UnitsConverter" sheet contains a 'units converter' to standardize all surface runoff and soil erosion measurements into the same unit (i.e., soil erosion in units of t/ha/yr or $g/m^2/hr$; runoff in units of mm/yr or mm/hr). The "CountryCode" sheet holds the international country code for the usage of Site_ID. The "Slope" sheet provides the converter of transforming slope from % to °. The "Quality_flag" sheet describes the quality control flag of measurements collected from papers (see **Table 4** for details). The "Meas_method" sheet describes soil erosion measurement methods reported in papers (see Table 1 for details). The "Biome" sheet describes different biome types. The "IGBP" sheet describes all 20 International Geosphere–Biosphere Programme (IGBP) (Townshend, 1992) vegetation types reported in papers. The "Manipulation" sheet includes description and comments about 17 manipulation types used in the *SoilErosionDB* (see **Table 5** for details). The "ReferenceList" sheet holds all reference details for all papers we compiled into the *SoilErosionDB*; and the "LiteratureSearch" sheets describes literature search details for the *SoilErosionDB,* such that users can reproduce the literature search results based on the description.

We read through each publication and compiled measurements and background information into *SoilErosionDB*. Currently we have collected and processed data from 124 papers that included measurements taken between 1980 and 2017 (**Figure 1b**). Each column in *SoilErosionDB* corresponds to either background information, surface runoff, soil erosion, or nutrient leaching indicator. When sites' location (latitude and longitude) was not reported, we estimated site coordinates according to the site name or the maps provided in the paper. For the soil erosion-related indicators, i.e., surface runoff, soil erosion, and nutrient leaching, data were either directly read from tables or digitized from figures. We used Data Thief (version III) (Flower et al., 2016) whenever we had to obtain the values from figures. Replications and standard deviation (SD) information were usually directly obtained from the original papers, however, sometimes confidence interval (CI), coefficient of variation (CV), or standard error (SE) was reported rather than SD, we calculated SD using equations 1-3 of (Jian et al., 2020a).

### 2.3 Surface runoff, soil erosion, and nutrient leaching measurements

The field, unit, and explanation about the surface runoff, soil erosion, and nutrient leaching measurements are presented in **Table 2**. It should be noted that *SoilErosionDB* has been designed to hold surface runoff, leaching and soil erosion measurements in terms of both annual amounts and instantaneous rates. However, nutrient leaching was organized in a different way, where the "*Leaching*" column holds values, the "*Leaching_unit*" holds the unit of measurement reported in the original paper, and the "*Leaching_type*" column records the leaching type reported in the original paper.

### 2.4 Background information



Background information (**Table 3**) includes descriptive data about sites and experimental design. Soil erosion measurement methods, quality control flag, and manipulation are further described in **Tables 1, Table 4, and Table 5**. Specifically, Table 1 describes 16 soil erosion measurement
methods reported in literature; Table 4 describes 10 quality control flags to help the developer record necessary information for quality control. Table 5 describes manipulation information, which is useful for the further analysis about how treatment affects surface runoff and soil erosion.

### 3 Technical validation

We carefully checked the data with the original paper to ensure the fidelity. We used the Mendeley
bibliography management software (https://www.mendeley.com) to ensure papers were not compiled into the database multiple times by different contributors. Each paper was first carefully read by the data collector, and any useful records were compiled into *SoilErosionDB*. Then a data quality checker compared the data in the database against the original paper. Specially, we paid attention to the methods sections, figures, and tables, where most of the surface runoff, soil erosion,
nutrient leaching, and background information were located.

In addition, we developed an R markdown file (ErosionDB_validation.Rmd in the Github repository) to examine the data quality of *SoilErosionDB*. The file was created using R Version 3.6.1 (R Core Team, 2020). For the latitude and longitude inputs, we plotted sites by individual country (currently a total of 22 countries, **Figure 2**), then compared the sites with that country's
boundaries to ensure that no sites fell outside. For any sites that appeared to be mislocated, we went back to the original paper and corrected the coordinates in the database. For all numeric columns in the *SoilErosionDB* (except "*Unique_ID*" and "*Study_number*"), we plotted histograms for each column, and checked whether extreme values were included in the database. **Figure 3** shows an example using the histograms of annual surface runoff and annual soil erosion.

### 150 4. Linkages to external data sources

Potentially important climate data (e.g., temperature and precipitation) are important factors affecting surface runoff and soil erosion; however, many papers did not report that information. Therefore, we linked the *SoilErosionDB* with a 0.5° × 0.5° resolution global climate data product (Willmott and C. J., 2000) to obtain annual temperature, mean annual temperature (MAT), annual
precipitation, and mean annual precipitation (MAP) based on site latitude and longitude. The MAT and MAP were calculated based on records between 1961 and 2015.

The results showed that temperature and precipitation data from the global climate dataset are highly correlated with that reported in the literature (**Figure 4**). Furthermore, we analyzed whether annual precipitation obtained from the external climate dataset can be used to explain annual soil



erosion variability in *SoilErosionDB*. We found that annual precipitation from the global climate
dataset can explain ~7% of variability in annual soil erosion (**Figure 5,** $R^2 = 0.07$, p = 0.01). We
presume that linking *SoilErosionDB* with other external data sources, e.g., leaf area index,
vegetation type, climate type, and soil properties, can lead to increased explanatory power for
spatial and temporal variability of soil erosion.

**5. Data and code availability**

The data and source code are available through GitHub
(https://github.com/jinshijian/*SoilErosionDB*) and Zenodo (Jian et al., 2020b)
(http://doi.org/10.5281/zenodo.4030875). The code is described in details with instructions for
users. Generally, a markdown file (*SoilErosionDB*.Rmd) were created, which generated all figures
(Figure 1 to Figure 5) and described the analysis for this study. All the data processing and data
visualization were conducted using R (version 3.6.1).

**6. Usage notes**

We suggest users download the data and code directly from Zenodo
(http://doi.org/10.5281/zenodo.4030875), as Zenodo provides DOI and generates the same results
for all users. Another advantage of using data and code from Zenodo is that it avoids any run errors
caused due to adding new measurements during database updating. On the other hand, the data
and code in the GitHub are for  development purposes. In addition, as new records are added to
the database, output results may differ from those generated using older versions, and may even
cause run errors. The users are encouraged to contact the *SoilErosionDB* development team before
using the data from Zenodo for analysis. We recommend that users contribute as a data quality
checker is a great first step to understand the data; with the provided R code, users could explore
the database as the code explained the analysis and the data in details.

**7. Future directions and contribution notes**

We have decided to share this work at this initial database development stage for two reasons: 1)
we want to receive feedback from the community about how to improve the data structure to ensure
optimal usage; 2) the large number of potentially relevant papers that have been or will be
published makes it important to expand the development team. Thus, we welcome and invite
scientists and data users who are interested in developing *SoilErosionDB* to download the dataset
and consider contributing published or unpublished data.  Our long-term goal is to update
*SoilErosionDB* by including measurements from newly published papers every year.





**Acknowledgements**

Xuan Du was supported by the Yangling Vocational & Technical College, under grant number: A2019009. Jinshi Jian was supported by the US Department of Energy, Office of Science, Biological and Environmental Research as part of the Terrestrial Ecosystem Sciences Program, under contract DE-AC05-76RL01830. Ryan Stewart was supported by the Virginia Agricultural Experiment Station and the Hatch Program of the National Institute of Food and Agriculture, U.S. Department of Agriculture. All the data and code to support this analysis can be found at:https://github.com/jinshijian/*SoilErosionDB* and http://doi.org/10.5281/zenodo.4030875. This database can be used for research, academic, individual, or commercial usage, and can be sold or repackaged without written permission.

**Author contributions**

Xuan Du and Jinshi Jian conceived the design of the data framework, compiled the data from papers to the *SoilErosionDB*. Xuan Du and Jinshi Jian wrote the manuscript, and all authors revised and approved the manuscript.

**Competing interests**

The authors declare no conflicts of interest.

**Figures and Tables**

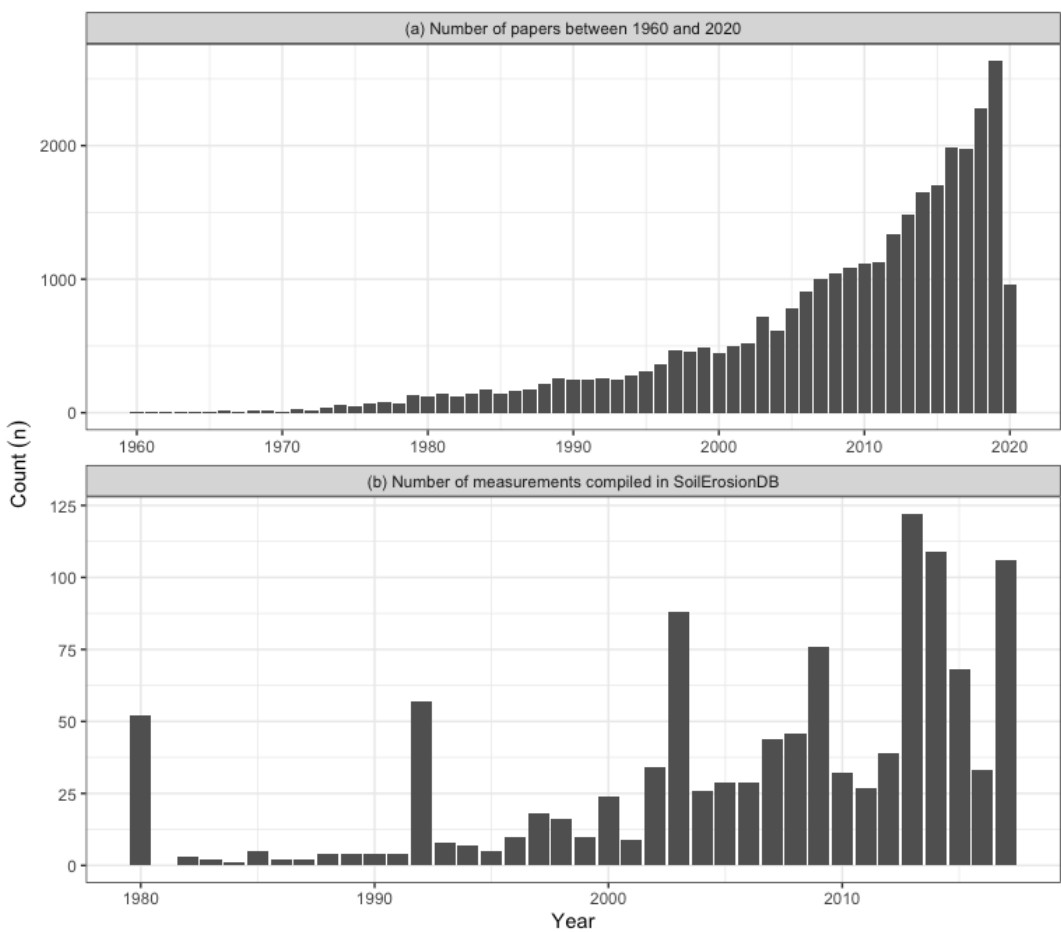

**Figure 1.** Summary of (a) studies published on the topics of surface runoff, soil erosion, and leaching between 1960 and 2020; and (b) temporal distribution of soil erosion measurements from 124 papers compiled into *SoilErosionDB*. The trend shows that more and more runoff and erosion related studies are published.

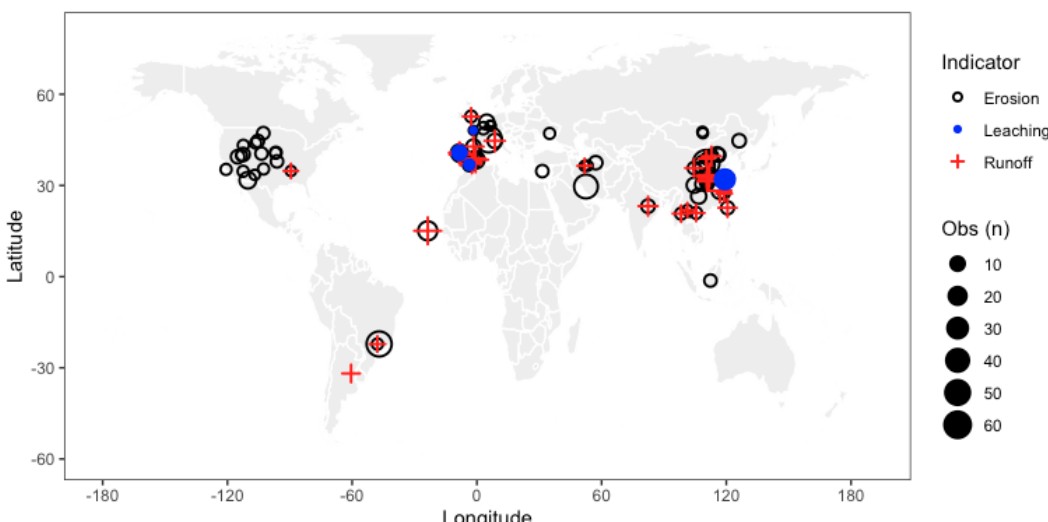

**Figure 2.** Spatial distribution of surface runoff, soil erosion, and leaching sites. The size of circles represents the sample size at each measurement site (i.e., bigger circles represent more data).



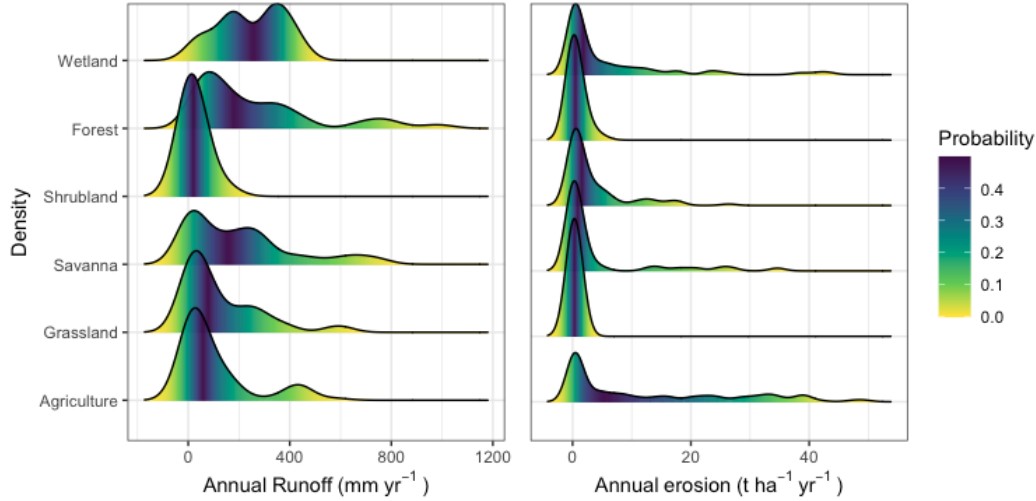

**Figure 3.** Distribution of surface runoff and soil erosion values in the global soil erosion database (*SoilErosionDB*). A few soil erosion data points with rates > 50 t ha$^{-1}$ yr$^{-1}$ were not shown.



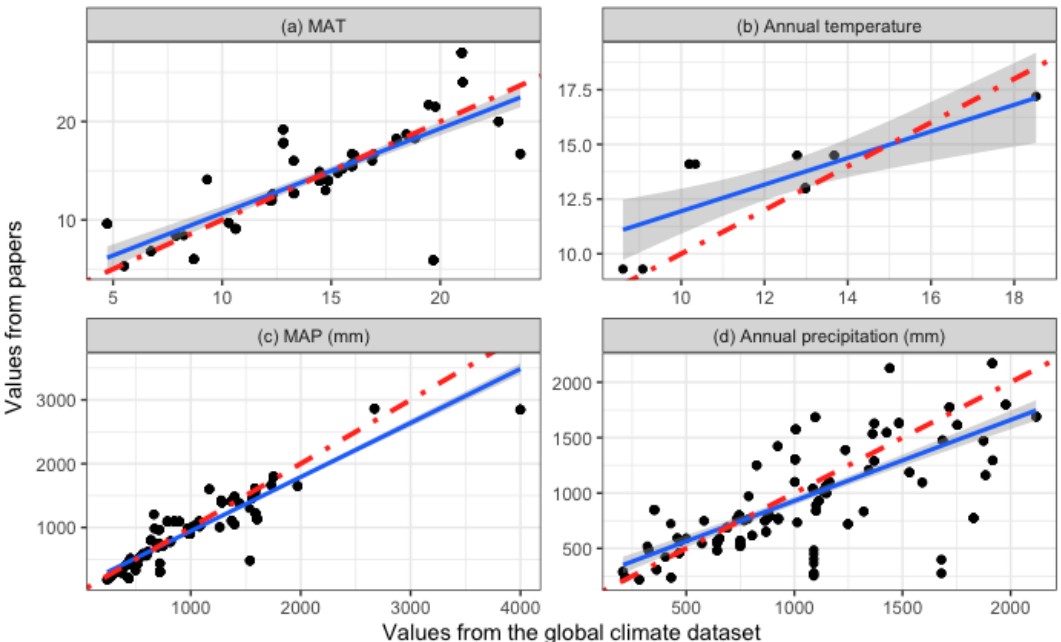

**Figure 4.** Relationship between (a and c) mean annual temperature/precipitation (between 1960 and 2017, MAT or MAP) reported in the papers and that from the global climate dataset (Willmott and C. J., 2000); and (b and d) annual temperature/precipitation obtained from papers vs. values obtained from the global climate dataset (Willmott and C. J., 2000). The linear regression lines
(solid blue) are very close to the 1:1 lines (dashed red), indicating a good agreement between air temperature (precipitation) reported in papers and values obtained from the global climate dataset. Note that the shaded regions around the regression lines indicate 95% confidence intervals.

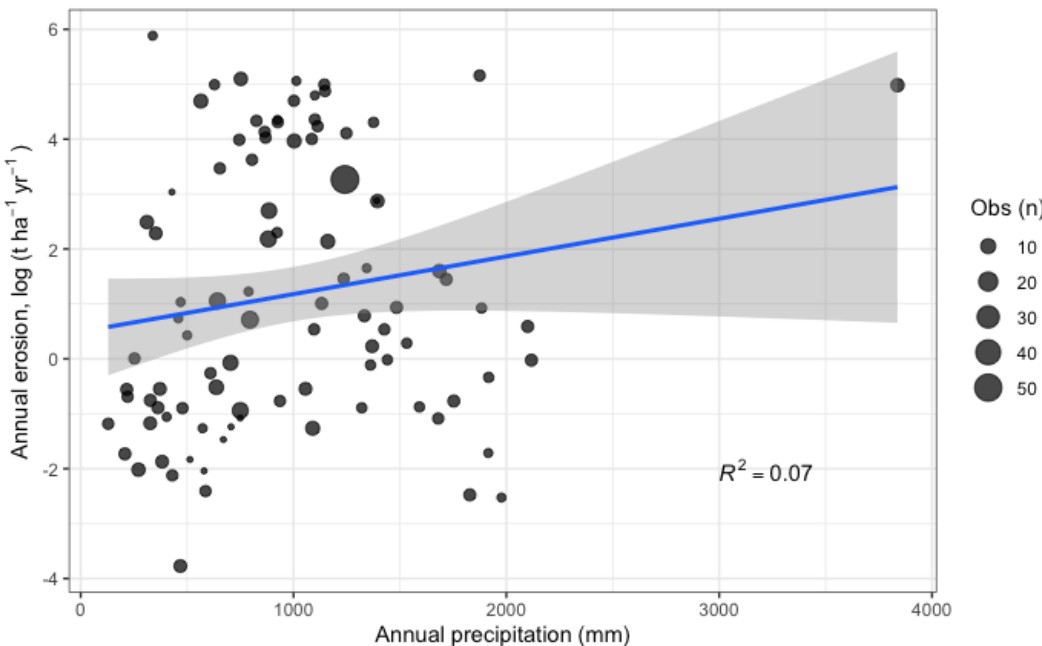

**Figure 5.** Relationship between annual soil erosion (log transferred, t ha$^{-1}$ yr$^{-1}$) and annual
precipitation (mm). The linear regression (solid blue line) showed that annual precipitation can
explain ~7% annual soil erosion variability ($R^2$ of the regression = 0.07 and p of slope = 0.01).
Note that the shaded region around the regression line indicates 95% confidence intervals. We
used a Cook's distance with threshold of 0.5 to detect any influential outliers, but no data points
were found to be outliers.





**Table 1.** Soil erosion measurement methods included in the global soil erosion database (*SoilErosionDB*).

| Method_Group | Method | Explanation |
|---|---|---|
| | Plot | E.g., standard runoff plot (20m × 5m) |
| | Morphology | Morphological transects |
| | Erosion pins | Erosion pins |
| | Profilemeters | Profilemeters |
| | Caesium 137 | Measuring soil erosion rate using Caesium137 |
| | Be7 | Measuring soil erosion rate using Be7 |
| | Catchment | Experimental catchment/watershed |
| | Reservoir surveys | Bathymetrical surveys of reservoirs |
| Field measurement | Field rainfall simulation | Rainfall simulation experiment in the field |
| Lab rainfall simulation | Lab rainfall simulation | Rainfall simulation experiment in the laboratory |
| | Model | Modelling, e.g., soil erosion rate calculated by the Revised Universal Soil Loss Equation (RUSLE) model (Foster et al., 2000; Nam et al., 2003) |
| | Remote sensing | Soil erosion estimated from remote sensing |
| | GIS | Soil erosion estimated based on GIS technology |
| Indirect methods | Topography | Topographic benchmarks related to vegetation |
| Other | Other | Measure methods not mentioned above |



**Table 2.** Description and categories of surface runoff and soil erosion related metrics in the global soil erosion database (SoilErosionDB).

| Field | Unit | Explanation |
|---|---|---|
| ER_annual | t/ha/yr | Annual soil erosion amount in unit of t/ha/yr |
| ER_annual_err | t/ha/yr | Annual soil erosion error (plot-to-plot) |
| ER_interann_err | t/ha/yr | Interannual error reported for annual soil erosion. This is sometimes reported in the article |
| ER_SD | t/ha/yr | Standard deviation (typically plot-to-plot) for annual soil erosion |
| ER_max | g/m$^2$/h | Maximum soil erosion rate |
| ER_maxday | No units | Day of year maximum soil erosion happen |
| ER_M_Area_h | g/m$^2$/hr | Soil erosion rate in unit of g/m$^2$/hr |
| ER_M_Volume | g/L | Soil erosion rate in unit of g/L |
| Runoff_annual | mm/yr | Annual Runoff amount in unit of mm/yr |
| Runoff_err | mm/yr | Error (typically plot-to-plot) for annual runoff |
| Runoff_SD | mm/yr | Annual runoff standard deviation (plot-to-plot) |
| Runoff_max | mm/h | Maximum runoff rate |
| Runoff_max_day | No units | Day of year when maximum runoff happen |
| Runoff_mm_h | mm/h | Runoff rate in unit of mm/h |
| Leaching | Units vary | Annual leaching amount or rate |
| Leaching_unit | Units vary | Annual leaching error (typically plot-to-plot) for |
| Leaching_SD | Units vary | Annual leaching standard deviation (plot-to-plot) |
| Leaching_type | No units | Leaching type (e.g., phosphorus, organic matter, organic carbon) |



**Table 3.** Description and categories of background fields in the global soil erosion database (*SoilErosionDB*).

| Field | Description | Comments |
|---|---|---|
| Unique_ID | Unique ID | A numeric unique ID number |
| Entry_date | Data entry date | When data was inputted, YYYY/MM/DD |
| Study_number | Paper identical ID | A numeric ID for each paper in *SoilErosionDB* |
| Author | First author's family name | |
| Duplicate_record | Duplicate sigh | If this study is a known duplicate, give the study number it repeated to |
| Quality_flag | Quality flag | Q0-Q10, see **Table 4** for details |
| Contributor | Initial of data collector | Initial of data collector, e.g., JJ for Jinshi Jian |
| Checker | Initial of data checker | Initial of data quality check person, e.g., XD for Xuan Du |
| Country | Data from which country | |
| Region | Data from which region | Usually, province or state |
| Site_name | Site name of the experiment | |
| Site_ID | Unique Site ID within a study | It is a combination of country code (see countrycode sheet), region code, and identify code (could be site name code, manipulation code, disturbance code, etc) |
| Paper_year | Paper published year | |
| Study_midyear | Year of data measured | |
| YearsOfData | Number of years | Data averaged from how many years, e.g., 3 for average from 3 years, usually it is 1 year |
| Annual_coverage | Annual coverage | 0-1, 0.01 means ≤ 1 day of data, 1 means covered at least a whole year, with at least 12 months of data |
| Latitude | Latitude of site | ° |
| Longitude | Longitude of site | ° |





| Elevation | Elevation of site | m |
|---|---|---|
| MAT | Mean annual temperature | °C |
| MAP | Mean annual precipitation | mm |
| Study_temp | Annual temperature | °C |
| Study_precip | Annual average precipitation | mm |
| MPET | Annual potential evapotranspiration | mm |
| Biome | Biome classification | Tropic, subtropic, temperate, Mediterranean, boreal, arctic etc. |
| STIR | Soil tillage intensity rating | |
| Manipulation_age | Years since manipulation | How many years since the manipulation |
| Manipulation | Manipulation | See **Table 5** for details |
| Manipulation_level | Management level | |
| Ecosystem_age | Year | Age of ecosystem |
| Species | Plant species of site | Latin name of dominant species |
| Leaf_habit | Leaf habit of site | Evergreen, deciduous, or mixed |
| Stage | Stage of site | Aggrading, mature, subjective |
| IGBP | IGBP classification | The International Geosphere–Biosphere Programme (IGBP) classification: https://climatedataguide.ucar.edu/climate-data/ceres-igbp-land-classification |
| Ecosystem_stage | Ecosystem stage | Ecosystem stage (natural, managed, unmanaged). Subjective |
| LAI | Leaf area index | Hemispheric (one-sided) if possible |
| Soil_family | Soil classification | Soil family, US classification if available |
| Soil_texture | Soil classification | Soil texture, US classification if available |
| Soil_sand | Sand percentage | % |





| Soil_silt | Silt percentage | % |
|---|---|---|
| Soil_clay | Clay percentage | % |
| Soil_rock | Rock fragment | % |
| Soil_BD | Soil bulk density | g cm$^{-3}$ |
| Soil_pH | Soil pH | Unitless |
| Soil_SAR | Sodic description | Unitless |
| Soil_C_% | Soil carbon concentration | % |
| Soil_C_stock | Soil carbon stock | g/m$^2$ |
| Soil_C_Depth | Soil carbon depth | cm |
| Soil_N | Soil nitrogen | % |
| Soil_CN | Soil carbon to nitrogen ratio | Unitless |
| Crust% | Soil crust | Unitless |
| Ksat | Soil saturated conductivity | cm/h |
| E | Revised Universal Soil Loss Equation (RUSLE) model rainfall factor, energy (Foster et al., 2000; Nam et al., 2003) | MJ ha$^{-1}$ year$^{-1}$ |
| I30 | RUSLE rainfall factor (Foster et al., 2000; Nam et al., 2003) | mm h$^{-1}$ |
| EI30_R | RUSLE rainfall factor (Foster et al., 2000; Nam et al., 2003) | E $\times$ I30, MJ mm h$^{-1}$ ha$^{-1}$ year$^{-1}$ |
| K | RUSLE soil erodibility factor (Foster et al., 2000; Nam et al., 2003) | t-ha-h ha-1 MJ$^{-1}$ mm$^{-1}$ |
| L | RUSLE slope length factor (Foster et al., 2000; Nam et al., 2003) | m |
| Slope | RUSLE slope factor (Foster et al., 2000; Nam et al., 2003) | ° |



| Plant_cover_C | RUSLE plant coverage factor (Foster et al., 2000; Nam et al., 2003) | % |
|---|---|---|
| P | RUSLE plant cropland management factor (Foster et al., 2000; Nam et al., 2003) | 0-1, unitless |
| Meas_method | Soil erosion measure method | See **Table 1** for details |
| Field_scale | Scale of experiment field area | Catchment, watershed, plot etc. |
| Field_area | Area of experiment field | $m^2$ |
| Measure_time | Time | When data was measured |
| Measure_interval | Frequency of measurement | How many minutes per measurement |
| Replication | Number of replications | |
| Rainfall_intensity | Rainfall intensity | Could be both in the field or in the lab, mm $h^{-1}$ |
| Rainfall_length | Rainfall length | How long the rainfall last, minute |
| Rainfall_amount | Rainfall amount | How much rainfall during the simulation, mm $h^{-1}$ |
| Notes | Notes | Other important notes about the data in this row |
| Data_source | Data source | Tips for where the data can be find in the paper |
| Other_comments | Other comments | Other important comments |



**Table 4.** Description for the quality flag field in the global soil erosion database (*SoilErosionDB*).

| Quality_flag | Description |
|---|---|
| Q0 | Default/none |
| Q1 | Estimated from figure |
| Q2 | Data not reported in the original paper, but could be found from another study |
| Q3 | Values estimated from a figure, with potential quality problem |
| Q4 | Potentially useful values in the future |
| Q5 | Values with potential problem |
| Q6 | Data need to double check with original authors |
| Q7 | Known problem |
| Q8 | Duplicate |
| Q9 | Inconsistency |
| Q10 | Lack of useful data |



**Table 5.** Description for the manipulation field in the global soil erosion database (*SoilErosionDB*).

| Manipulation Group | Manipulation | Comments |
|---|---|---|
| Control | Control | default, if no manipulation can be identified |
| Changes in precipitation | Precipitation pattern change | e.g., same precipitation amount as control, but different intensity and duration time |
| | Precipitation amount change | More or less precipitation comparing with control |
| Fertilization | Fertilization | e.g., N addition, slurry addition, compost |
| Agriculture cultivation | Conventional tillage | Traditional tillage (using plough, destroy soil structure a lot) |
| | Cover crop | One type of conservation management |
| | No-till | One type of conservation management |
| | Reduced tillage | One type of conservation management |
| | Mulch | One type of conservation management |
| | Conservation tillage | Other conservation management other than cover crop, no-till, reduced tillage, mulch, and ridge tillage |
| Pollution | Pollution | Human activities related prolusion, such as acid rain |
| PAM | PAM | Polyacrylamide application |
| Fire | Fire | Disturbance by wildfire or artificial fire |
| Multiple factors | Multiple factors | Interactive and relative effects of two or more than two manipulations |
| Others | Others | Other treatment not included above |

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
