# Peer review of "SoilErosionDB: A global database for surface runoff and soil erosion evaluation"

_Earth System Science Data, 2020_

## Referee Comment (RC1) · Karl Auerswald (Referee) · 24 Feb 2021

Soil erosion is a process of high temporal variability where single events may differ by several orders of magnitude. Given that the number of events per year is limited, long periods of measurement are required for obtaining reliable averages. These periods of time are usually much longer than what can be reasonably financed and maintained. Most studies on soil erosion thus suffer from too short measuring periods. Furthermore, erosion is highly variable in space. This calls for many measuring sites which also cannot be afforded. Both problems may be attenuated by combining the data of many studies. It is thus applaudable that Jian et al. took the effort to compile such a database, which presently contains data from 124 publications but which is intended to grow in the future. The key property of such a database, however, is not the number of

studies included but the reliability of the data, which is difficult to achieve for different reasons originating already from the individual studies (e. g. inconsistent and incomplete variables, use of different units, differing spatial and temporal resolution) but also from insufficiencies by the compilers that often result from the tradeoff between the number of studies and the time remaining for the individual study. I will assess the quality of the database below but first I will assess the manuscript.

The title of the manuscript is rather unspecific as many types of erosion exist (to name some: water erosion, wind erosion, tillage erosion, coastal erosion, genetic erosion, bone erosion and many more). Neither the title nor the abstract tells, which one is addressed. The term runoff lets me speculate that water erosion is addressed but early in the Background the dust bowl is mentioned pointing to wind erosion. The Science Direct search (erosion, runoff) implies that all types of erosion including those that have no relation to soil are meant (note: this is not clear because in one place the authors wrote that they searched for 'erosion and runoff' while in two other places including the database they searched for 'erosion, runoff'). From the text and the database itself I got the impression that the authors looked for sheet and rill erosion (but not for gully erosion). The authors have to be precise!

The manuscript does not follow the structure of scientific manuscripts with Introduction, Material and methods, Results and Discussion but these parts are intermingled. This makes is difficult locating a specific information and likely is the reason why some information is repeated several times despite the short length of the manuscript (e. g. the address of the database is given in four places).

The introduction is called background and briefly describes erosion (many types like that caused by avalanches or animal trampling, which doesn't help to clarify the topic). Surprisingly, it does not summarise the existing data collections on sheet and rill erosion that already exist and could have easily incorporated into the new database. Likely this would also have helped to identify some problems associated with data compilation. Even without reading and reporting the existing studies, the authors conclude out

of the blue that there was no successful effort to compile data from several studies into a single and coherent dataset.

I appreciate that the authors attempted to make the Chinese data available. However, from the database I learned that the authors had considerable problems of correctly understanding European studies. Russian studies were completely missing. I would recommend expanding the consortium to include a wider regional experience, which is indispensable for correctly interpreting the data. Also, given the multitude of sciences that work together and which are required (e. g. geomorphology, meteorology, agronomy in very different regional settings, soil science, vegetation science), I recommend to widen the consortium regarding the sciences as well. For instance, from the database it became clear that the authors seem not to be aware of the difference of SOM and SOC and that soil carbon is an ambiguous term because it may be organic, mineral or total carbon. These differences would be well aware to any soil scientist.

The authors decided to include data without any quality control (e.g., also papers without peer review). This invalidates the entire database even when excellent studies are included. Bad data don't become better when many of them are compiled or when they are mixed with good data but they spoil the good data.

Assessment of database: Due to my specific knowledge of the German situation, I looked at the entries for Germany. Only two studies are incorporated. This is surprising because several compilations and databases exist for Germany, which compile already the data of many studies on natural-rain plots (27 studies covering 1076 plot years compiled by Auerswald et al. 2009), rainfall simulations studies (726 simulations compiled by Fiener et al. 2011) and small-watershed data (112 watershed years with daily resolution compiled by Fiener et al. 2019).

Which two studies were selected for Germany is not clear because the database does not report the sources. Hence there is no chance to complete the data, identify errors or look for an interpretation of results that are not self-explaining. It appears indispensable

that the database reports the full source information. This is already necessary to acknowledge the hard work of the experimenters. The authors of the database would also not want their database to be used without acknowledging their work.

I concentrate on the second German publication in the database, which likely is Rodrigo Comino et al. (2019); at least the truncated name of the first author in the database, the year and some data agree and support this assumption. I found no other publication that would fit to these data. The database reports two sites while the publication reports only the first one. The reason for the discrepancy remains unclear.

MAP and MAT are given in the publication but the database assigns these as study precipitation and study temperature (= wrong columns).

The database reports the biom Mediterranean, which does not exist in Germany.

The publication mentions only nine rainfall simulations while the database reports twelve replications at five sites (= 60 rainfall simulations). Only the values of the first data set agree with the publication. The origin of the other four datasets is unclear.

The publication reports SOM. The identical value is reported in the database but as SOC.

Intensity and duration of the rain is given but not the amount. Why? This makes data selection and retrieval trickier than necessary.

The variable names and the units of many variables in the database are unclear. Some (very few) examples:

Is ER_annual a multi-year mean or an individual year?

What is an interannual error (standard error, standard deviation...)?

What is ER_max? From the different unit that is used here, I expect that this is on a shorter time scale than years but which? From ER_max day I speculate that this may be the maximum of daily soil loss. All variables need much better description.

ER_M_Area_h is an especially confusing variable because also ER_annual is the mass lost per area and time. Is this variable only obtained by unit conversion or is it something different?

Leaching? What is meant with leaching? Nitrate leaching to groundwater? This was already mysterious in the manuscript.

MAP and MPET carry the unit mm/yr!

From the unit of Study_precip I speculate that this is event precipitation. Or is it annual precipitation and the temporal unit was wrongly omitted as for MAP. In any case, the database user should not be forced to speculate.

Stage: is subjective also a stage?

LAI: what does 'if possible' mean?

Sand and other variables: the unit % is meaningless unless the base is given (e.g. percent of volume or of dry or wet weight, percent of bulk soil or of fine earth fraction). The definition of the soil particle sizes differs among countries. Which definition is used here? Is it consistently used? Soil carbon concentration: which carbon identity?

These were just some striking examples. Virtually the description of all variables has to be improved. I wonder how data compilation was done with such imprecise definition of variables.

In summary: I highly applaud the effort by the authors but given the many deficits, particularly in the database itself, they should better start from the scratch.

References: Auerswald, K., Fiener, P., Dikau, R. (2009): Rates of sheet and rill erosion in Germany – a meta-analysis. Geomorphology 111: 182–193. http://dx.doi.org/10.1016/j.geomorph.2009.04.018

Fiener, P.; Seibert, S.; Auerswald, K. (2011): A compilation and meta-analysis of rainfall simulation data on arable soils. Journal of Hydrology 409: 395–406,

http://dx.doi.org/10.1016/j.jhydrol.2011.08.034

Fiener P., Wilken F., Auerswald K. (2019): Filling the gap between plot and landscape scale – eight years of soil erosion monitoring in 14 adjacent watersheds under soil conservation at Scheyern, Southern Germany. Advances in Geosciences 48, 31–48, https://doi.org/10.5194/adgeo-48-31-2019

Rodrigo Comino J, Iserloh T, Morvan X, Malam Issa O, Naisse C, Keesstra SD, Cerdà A, Prosdocimi M, Arnáez J, Lasanta T, Ramos MC, Marqués MJ, Ruiz Colmenero M, Bienes R, Ruiz Sinoga JD, Seeger M, Ries JB. Soil Erosion Processes in European Vineyards: A Qualitative Comparison of Rainfall Simulation Measurements in Germany, Spain and France. Hydrology. 2016; 3(1):6. https://doi.org/10.3390/hydrology3010006

———————————————

---

## Referee Comment (RC2) · Anonymous Referee #2 · 12 Apr 2021

This reviewer totally agreed with Prof. Dr. Karl Auerswald's comments. A global database for surface runoff and soil erosion is very valuable for research community in hydrology and soil erosion. This reviewer was very interested in the database and downloaded SoilErosionDB. However, I worried about its reliability and utility. Soil erosion is a very complex process as it relates various aspects and factors including climate, soil, topography, biology and human activities, and it covers different spatial and temporal scales. The results from different scales can not compare directly, which should be paid more attention to. This reviewer would suggest the authors start from a small topic, such as the collection of observations for the runoff and soil erosion of field plots, which are very important for the development and calibration of soil erosion models. In addition, before the release of the database, strict quality control and analysis

on the dataset are necessary.

---

## Author Comment (AC1) · 7 May 2021

We greatly appreciate the second anonymous referee #2 for their helpful and insightful comments. Please see below our responses:

Comments: This reviewer totally agreed with Prof. Dr. Karl Auerswald's comments. A global database for surface runoff and soil erosion is very valuable for research community in hydrology and soil erosion. This reviewer was very interested in the database and downloaded SoilErosionDB.

Response: We thank you for your interest in our database.

Comments: However, I worried about its reliability and utility. Soil erosion is a very

complex process as it relates various aspects and factors including climate, soil, to-pography, biology and human activities, and it covers different spatial and temporal scales. The results from different scales can not compare directly, which should be paid more attention to. This reviewer would suggest the authors start from a small topic, such as the collection of observations for the runoff and soil erosion of field plots, which are very important for the development and calibration of soil erosion models.

Response: We agree and will only keep those measurements from field plots experiments in our database.

Comments: In addition, before the release of the database, strict quality control and analysis on the dataset are necessary.

Response: We will do a careful and strict quality control and analysis on the dataset before releasing our new version. Please see the specific responses to Reviewer #1 for more details.

---

## Author Comment (AC2) · 7 May 2021

*Dear Professor Auerswald,*

*We thank you for your insightful comments on our manuscript "SoilErosionDB: A global database for surface runoff and soil erosion evaluation" submitted to ESSD. We appreciate your time and efforts to evaluate both the quality of the manuscript as well as the database, your comments are very helpful for us to improve the quality of both the manuscript and the dataset. According to your comments and comments from the second referee, we will carefully re-design our database, implement more stringent quality control and quality inspection to the database. With those efforts, we hope the quality of the database and the manuscript will be greatly improved in the future.*

*Thank you for your time and your valuable comments.*

*Sincerely,*

*Jinshi Jian*

Soil erosion is a process of high temporal variability where single events may differ by several orders of magnitude. Given that the number of events per year is limited, long periods of measurement are required for obtaining reliable averages. These periods of time are usually much longer than what can be reasonably financed and maintained. Most studies on soil erosion thus suffer from too short measuring periods. Furthermore, erosion is highly variable in space. This calls for many measuring sites which also cannot be afforded. Both problems may be attenuated by combining the data of many studies. It is thus applaudable that Jian et al. took the effort to compile such a database, which presently contains data from 124 publications but which is intended to grow in the future. The key property of such a database, however, is not the number of studies included but the reliability of the data, which is difficult to achieve for different reasons originating already from the individual studies (e. g. inconsistent and incomplete variables, use of different units, differing spatial and temporal resolution) but also from insufficiencies by the compilers that often result from the tradeoff between the number of studies and the time remaining for the individual study. I will assess the quality of the database below but first I will assess the manuscript.

*Response: we thank you for your overall positive opinion on this work. We agree with your opinion that "the key property of such a database is not the number of studies but the reliability of the data". This actually was our original intention when submitting our manuscript to ESSD at the beginning of this work: we want more comments and suggestions on how to improve this database before we proceeded too far, and we are grateful for the high quality feedbacks from two referees. Our preprint at Researchgate received significant reads and feedback as well-for example, Dr. Evans contacted us and wants to contribute to expanding and improving SoilErosionDB by combining it with his soil erosion dataset (from sites across 38 countries); for more details about Dr. Evans' dataset please see* https://iopscience.iop.org/article/10.1088/1748-9326/aba2fd. *We believe that the quality of the database will greatly benefit from these comments and contributions.*

The title of the manuscript is rather unspecific as many types of erosion exist (to name some: water erosion, wind erosion, tillage erosion, coastal erosion, genetic erosion, bone erosion and

many more).  Neither the title nor the abstract tells, which one is addressed. The term runoff lets me speculate that water erosion is addressed but early in the Background the dust bowl is mentioned pointing to wind erosion.  The Science Direct search (erosion, runoff) implies that all types of erosion including those that have no relation to soil are meant (note: this is not clear because in one place the authors wrote that they searched for 'erosion and runoff' while in two other places including the database they searched for 'erosion, runoff').  From the text and the database itself I got the impression that the authors looked for sheet and rill erosion (but not for gully erosion). The authors have to be precise!

*Response: Thank you for pointing out this lack of detail; here we focus on water erosion, more specifically, sheet and rill erosion but not gully erosion. We used key words 'erosion, runoff' in https://www.sciencedirect.com/ to find published papers on erosion. Papers related to all types of erosion were downloaded, but only those that met the conditions (sheet and rill erosion) upon screening were kept. We will clarify this criterion in the revision, including specifying the types of erosion in the abstract.*

The manuscript does not follow the structure of scientific manuscripts with Introduction, Material and methods, Results and Discussion but these parts are intermingled. This makes is difficult locating a specific information and likely is the reason why some information is repeated several times despite the short length of the manuscript (e. g. the address of the database is given in four places).

*Response: We will re-write the manuscript following the instruction and remove all repeated material.*

The introduction is called background and briefly describes erosion (many types like that caused by avalanches or animal trampling, which doesn't help to clarify the topic). Surprisingly, it does not summarise the existing data collections on sheet and rill erosion that already exist and could have easily incorporated into the new database. Likely this would also have helped to identify some problems associated with data compilation. Even without reading and reporting the existing studies, the authors conclude out of the blue that there was no successful effort to compile data from several studies into a single and coherent dataset.

*Response: We will carefully investigate the existing data collections on sheet and rill erosion that already exist, such as Erosion Plot Database (EPD) for Loess Plateau by Zhao et al (2016), data from all available studies on soil loss under natural rainfall in Germany compiled by Auerswald et al (2009), and a compilation of 4285 plot-based gross erosion rates representing 10,030 plot years amassed from 240 studies across 38 countries by Evans et al (2020). We will incorporate data from those datasets into the SoilErosionDB. Dr. Evans already agreed to contribute as a coauthor, and we will contact Dr. Zhao and this reviewer (Dr. Auerswald) and ask if they would like to contribute as coauthors. Given that, the introduction will be rewritten.*

*Zhao, Jianlin, et al. "Moderate topsoil erosion rates constrain the magnitude of the erosion-induced carbon sink and agricultural productivity losses on the Chinese Loess Plateau." Biogeosciences 13.16 (2016): 4735-4750.*

*Auerswald, Karl, Peter Fiener, and R. Dikau. "Rates of sheet and rill erosion in Germany—A meta-analysis." Geomorphology 111.3-4 (2009): 182-193.*

*Evans, D. L., et al. "Soil lifespans and how they can be extended by land use and management change." Environmental Research Letters 15.9 (2020): 0940b2.*

I appreciate that the authors attempted to make the Chinese data available. However, from the database I learned that the authors had considerable problems of correctly understanding European studies.  Russian studies were completely missing.  I would recommend expanding the consortium to include a wider regional experience, which is indispensable for correctly interpreting the data.  Also, given the multitude of sciences that work together and which are required (e. g. geomorphology, meteorology, agronomy in very different regional settings, soil science, vegetation science), I recommend to widen the consortium regarding the sciences as well. For instance, from the database it became clear that the authors seem not to be aware of the difference of SOM and SOC and that soil carbon is an ambiguous term because it may be organic, mineral or total carbon. These differences would be well aware to any soil scientist.

*Response: We agree with that our terminology and descriptions of soil carbon terms were insufficient. We will revise our manuscript and database to be more accurate. In terms of expanding the consortium and bringing scientists from different backgrounds, that is a great suggestion (see detailed responses above).*

The authors decided to include data without any quality control (e.g., also papers without peer review). This invalidates the entire database even when excellent studies are included. Bad data don't become better when many of them are compiled or when they are mixed with good data but they spoil the good data.

*Response: We will introduce quality control to the data sources. We will add a quality flag (peer reviewed or not) to identify whether the data source is peer reviewed or unpublished data, we make this decision is because 1) there are lots of data not published; 2) unpublished data is important in some analysis such as meta-analysis to avoid the publication bias. With this quality flag, users can make their own decision whether include the unpublished data or not.*

Assessment of database:  Due to my specific knowledge of the German situation, I looked at the entries for Germany.  Only two studies are incorporated.  This is surprising because several compilations and databases exist for Germany, which compile already the data of many studies on natural-rain plots (27 studies covering 1076 plot years compiled by Auerswald et al. 2009), rainfall simulations studies (726 simulations compiled by Fiener et al. 2011) and small-watershed data (112 watershed years with daily resolution compiled by Fiener et al. 2019).

*Response: Thank you for your suggestion. We will integrate data from Auerswald et al. 2009 into our database in the revision. However, following the second referee's suggestion, we decided to only include field plots measured data in this database, so data from Fiener et al. 2011 and Fiener et al. 2019 will not be included in the SoilErosionDB.*

Which two studies were selected for Germany is not clear because the database does not report the sources. Hence there is no chance to complete the data, identify errors or look for an

interpretation of results that are not self-explaining. It appears indispensable that the database reports the full source information. This is already necessary to acknowledge the hard work of the experimenters. The authors of the database would also not want their database to be used without acknowledging their work.

*Response: Thank you for pointing out this lack of sufficient information; the reference list will be updated in the revision. Specifically, the German studies were:*

*Rodrigo Comino, J. et al. Quantitative comparison of initial soil erosion processes and runoff generation in Spanish and German vineyards. Sci. Total Environ. 565, 1165–1174 (2016).*

*Khosh Bin Ghomash, S., Caviedes-Voullieme, D. & Hinz, C. Effects of erosion-induced changes to topography on runoff dynamics. J. Hydrol. 573, 811–828 (2019).*

I concentrate on the second German publication in the database, which likely is Rodrigo Comino et al. (2019); at least the truncated name of the first author in the database, the year and some data agree and support this assumption. I found no other publication that would fit to these data. The database reports two sites while the publication reports only the first one. The reason for the discrepancy remains unclear.

*As stated in our response above, Professor Auerswald assumed a different study (Rodrigo et al., 2019) than the one we actually used (Rodrigo et al., 2016). This source of confusion was caused by our incomplete references, yet we believe that the difference is why Professor Karl Auerswald identified many disagreements in the database. Please see our response to comments below based on Rodrigo et al. (2016).*

MAP and MAT are given in the publication but the database assigns these as study precipitation and study temperature (= wrong columns).

*Response: Thank you for your careful checking. We have included MAT, MAP, study_temp, and study_precip as four columns in the database. MAT and MAP are columns for mean annual temperature and mean annual precipitation, while study_temp and study_precip are columns for annual temperature and annual precipitation in the study. Table 1 reports annual temperature and annual precipitation, so those values are in the correct columns.*

| Geology | Cretaceous limestones and Tertiary Marly deposits | Metamorphized schits and quartzites | Devonian greywackes, slates and quartzites | Devonian greywackes, slates and quartzites |
|---|---|---|---|---|
| T° ($\bar{x}$)[1] | 14.2 | 17.2 | 9.3 | 14.1 |
| T° (max_$\bar{x}$)[2] | 25 | 24.9 | 17.6 | 24.6 |
| T° (min_$\bar{x}$)[3] | 9.2 | 11.3 | 1.5 | 4.8 |
| Pp ($\bar{x}$total)[4] | 420 | 520 | 765 | 749 |
| Pp(max_$\bar{x}$)[5] | 42 | 83.6 | 71.2 | jinshi9.3 |
| Pp (min_$\bar{x}$)[6] | 5 | 2.9 | 50.6 | 37.1 |

T°($\bar{x}$)[1] = Annual average temperatures; T° (max_$\bar{x}$)[2] = Maximal monthly average temperatures; T° (min_$\bar{x}$)[3] = Minimal monthly average temperatures; Pp ($\bar{x}$total)[4] = Average of annual rainfall depth; Pp (max_$\bar{x}$)[5] = Maximal monthly average rainfall depth; Pp (min_$\bar{x}$)[6] = Minimal monthly average rainfall depth; TOC[7] = Total organic content.

The database reports the biom Mediterranean, which does not exist in Germany.

*Response: Thank you for pointing out this. We will delete biome, and instead provide code to link SoilErosionDB with the climate Köppen climate classification so the user can get climate information from there.*

The publication mentions only nine rainfall simulations while the database reports twelve replications at five sites (= 60 rainfall simulations). Only the values of the first data set agree with the publication. The origin of the other four datasets is unclear.

*Response: There were 83 simulations in 7 sites, with approximately 12 replications, please see the screenshot below:*

**2.3. Procedures and evaluation of the comparison of rainfall simulation experiments**

Test duration per rainfall simulation was 30 min. Most of the rainfall simulations were carried out during 2013 and 2015, except the few made in 2008 at Waldrach. 83 rainfall simulations were performed in total (Table 2).

The publication reports SOM. The identical value is reported in the database but as SOC.

*Response: Soil total organic carbon values were reported in Table 1.*

**Table 1**
Soil, climatic and geological context.

| Test area | Moixent (M) | Almáchar (A) | Waldrach (W) | | Kanzem (K) | | |
|---|---|---|---|---|---|---|---|
| Type of vineyard | Conventional | Conventional | Conventionalold (WO) | Conventionalyoung (WY) | Conventionalold (KCO) | Ecologicalold (KEO) | Ecologicalyoung (KEY) |
| Clay (%) | 8 | 5.6 | 9.4 | 8.9 | 10.6 | 8.1 | 9.69 |
| Silt (%) | 32 | 72.2 | 64.7 | 64.3 | 40.4 | 31 | 35 |
| Sand (%) | 60 | 22.2 | 26 | 26.8 | 49 | 60.9 | 55.31 |
| TOC[7] (%) | 1.01 | 3.1 | 6.1 | 7.9 | 9 | 5.4 | 6.7 |
| pH | 7.8 | 7.1 | 7.2 | 6.5 | 7.5 | 7.2 | 7.3 |
| Coordinates | 38.7833 N; 0.87E | 36.8 N; −4.2167 W | 49.7418 N; 6.7524E | | 49.6667 N; 6.5756E | | |

Intensity and duration of the rain is given but not the amount. Why? This makes data selection and retrieval trickier than necessary.

*Response: Thank you for the suggestion. That's right, the rainfall volume can be calculated as rainfall intensity × duration. We will do the calculation in the revision.*

The variable names and the units of many variables in the database are unclear. Some (very few) examples:

Is ER_annual a multi-year mean or an individual year?

*Response: If 'YearOfData' (column O) is 1, then ER_annual is based on an individual year, but it is a multi-year mean if 'YearOfData' > 1. We will clarify this point in the database documentation.*

What is an interannual error (standard error, standard deviation...)?

*Response: We meant standard error, and will clarify this information in the revision.*

What is ER_max? From the different unit that is used here, I expect that this is on a shorter time scale than years but which? From ER_max day I speculate that this may be the maximum of daily soil loss. All variables need much better description.

*Response: That's right, it is in a different unit ($g/m^2/h$). You are correct, ER_max and ER_max_day are used to record the maximum of daily soil loss and the date it happened. From the database, we see that no papers have reported this so far, and we will discuss whether to keep these columns.*

ER_M_Area_h is an especially confusing variable because also ER_annual is the mass lost per area and time. Is this variable only obtained by unit conversion or is it something different?

*Response: That's true, and we will change this column name to "ER_g_m2_hr".*

Leaching? What is meant with leaching? Nitrate leaching to groundwater? This was already mysterious in the manuscript.

*Response: Nutrients in groundwater are reported in different ways, such as Nitrate and Phosphorus, those may also reported in different unit, so we decide organize this differently: have a Leaching_unit and a Leaching_type column to describe it. But we notice this may cause confusion to users, we will discuss and may only including Nitrate (as this is mainly reported in all the publication in the database) in the revision.*

MAP and MPET carry the unit mm/yr!

*Response: Thank you—we will change the unit of MAP and MPET, and will also check units for all other columns.*

From the unit of Study_precip I speculate that this is event precipitation. Or is it annual precipitation and the temporal unit was wrongly omitted as for MAP. In any case, the database user should not be forced to speculate.

*Response: study_temp and study_precip are columns for annual temperature and annual precipitation in the study. In order to avoid any confusion, we will change the column names "study_temp" and "study_precip" to "Annual_temp" and "Annual_precip".*

Stage: is subjective also a stage?

*Response: Subjective is not a stage, rather we mean this information is very subjective. We will explain it to avoid the confusion.*

LAI: what does 'if possible' mean?

*Response: We meant we will record the information 'if available'. We will change this term into 'if available' in the revision.*

Sand and other variables: the unit % is meaningless unless the base is given (e.g. percent of volume or of dry or wet weight, percent of bulk soil or of fine earth fraction). The definition of the soil particle sizes differs among countries. Which definition is used here? Is it consistently used? Soil carbon concentration: which carbon identity?

*Response: This is a great question. The "%" means percent of dry weight using the American definition (sand - 2.0 and 0.05 mm; silt - 0.05 mm and 0.002 mm; and clay - less than 0.002 mm). We will explain this in the revision, and only data that meet this requirement will be included in the database.*

These were just some striking examples. Virtually the description of all variables has to be improved. I wonder how data compilation was done with such imprecise definition of variables.

*Response: We will carefully check and clarify the description for all variables in the revision.*

In summary: I highly applaud the effort by the authors but given the many deficits, particularly in the database itself, they should better start from the scratch.

*Response: Thank you for your insightful comments and suggestions. We will re-design this database and try our best to improve the quality of the database.*